# Effect of Two-Step Formosolv Fractionation on the Structural Properties and Antioxidant Activity of Lignin

**DOI:** 10.3390/molecules27092905

**Published:** 2022-05-02

**Authors:** Xiaoxia Duan, Xueke Wang, Ao Huang, Guijiang Liu, Yun Liu

**Affiliations:** 1Beijing Key Laboratory of Bioprocess, College of Life Science and Technology, Beijing University of Chemical Technology, Beijing 100029, China; duanxiaoxia@zeqiao.com (X.D.); 2020201165@buct.edu.cn (X.W.); 2021210745@mail.buct.edu.cn (A.H.); 2021210746@mail.buct.edu.cn (G.L.); 2Beijing Zest Bridge Media Technology Inc., Beijing 100176, China

**Keywords:** lignin, formosolv fractionation, depolymerization, structural properties, antioxidant activity

## Abstract

The formosolv fractionation process has been demonstrated to be an effective approach toward lignin recovery as an antioxidant from lignocellulosic biomass. In this study, four lignin fractions, FL-88%, FSL-70%, FIL-70% and FL-EtAc, were isolated from *Phragmites australis* biomass through two-step formosolv fractionation (88% formic acid delignification followed by 70% aqueous formic acid fractionation). To better understand the structural properties of the lignin obtained from this fractionation process, four isolated lignins were successfully characterized by gel permeation chromatography (GPC), Fourier transform infrared (FT-IR), two-dimensional heteronuclear single quantum coherence nuclear magnetic resonance (2D-HSQC NMR) spectroscopy, thermogravimetric analysis (TGA) and gas chromatograph-mass spectroscopy (GC/MS). It was found that lignin depolymerization via β-*O*-4 cleavage occurred via a formylation, elimination and hydrolysis mechanism, accompanied by a competitive condensation reaction. Noteworthily, two-step formosolv fractionation can produce specific lignin fractions with different ABTS and DPPH radical scavenging activities. The FL-EtAc fraction with low molecular weight (M_w_ = 2748 Da) and good homogeneity (PDI = 1.5) showed excellent antioxidant activity, compared with the other three isolated lignin fractions, even equal to that of commercial antioxidant BHT at the same concentration of 2.0 mg·mL^−1^. These findings are of great help for specific lignin from biomass as a natural antioxidant in the future.

## 1. Introduction

Lignin is a naturally renewable aromatic polymer consisting of guaiacyl (G), syringy (S), and *p*-hydroxybenzyl (H) subunits [1]. It has been extensively considered that lignin endows potential promising antioxidant activity as a versatile candidate applied in biomaterials [2]. The antioxidant activity of lignin is significantly dependent on lignin structure and heterogeneity, such as hydroxyl groups, methoxy groups and molecular weight [2,3]. However, the structural properties of technical lignin can be severely modified during biomass pretreatment, which highly restricts lignin bioactivities [4]. Therefore, fractionation processes to obtain specific lignin fractions became very challenging but mandatory.

Lots of fractionation processes related to specific technical lignin from biomass have been investigated in recent years. The most common ones to obtain technical lignin include organic solvent extraction [4,5], selective acid precipitation [3,6], and ultrafiltration techniques [7,8]. Among the abovementioned processes, lignin obtained from biomass by formic acid (FA) fractionation has been demonstrated to be a facile and environmentally friend approach. The merits of using FA as a fractionation solution are: (1) FA is the renewable solution, which can be derived from biomass; (2) FA has a low boiling point of 100.6 °C, which can be easily recycled for usage; (3) FA has a high lignin dissolving capacity. In our previous work, the isolated technical lignin from *Eucalyptus* biomass by 70% aqueous FA fractionation was investigated, and 73.18% of technical lignin with a molecular mass weight of 2582 g·mol^−1^ was obtained with depolymerization to some content [9]. Shao and co-workers pointed out that lignin depolymerization was accompanied by a condensation reaction during the FA fractionation process [10]. Oregui-Bengoechea and co-workers unraveled the distinctive role of FA for the conversion of lignin into bio-oil and a formylation elimination-hydrogenolysis mechanism for the FA-aided lignin conversion was proposed [11].

To avoid condensation reactions and enhance the yield of the lignin monomers, Rahimi and co-workers considered that the depolymerization of oxidized lignin in an aqueous FA solution could achieve a high yield of aromatic monomers, the content of which was more than 60 wt.% [12]. In order to avoid the formation of interunit carbon–carbon bonds within depolymerized lignin fractions, Shuai and co-workers reported that adding formaldehyde during biomass fractionation could avoid lignin repolymerization and produce aromatic monomers at near theoretical yields during subsequent hydrogenolysis [13]. In regards to the FA fractionation process, previous works have confirmed that FA does not merely act as a solvent reagent but also seems to react with lignin to generate formate esters [9,10]. However, Halleraker and co-workers convinced that carbon from FA should be incorporated into carboxylic acids derived from the depolymerized lignin bio-oil, not an outcome of a formylation reaction between lignin and FA [14].

Although a lot of reports on FA fractionation of lignocellulosic biomass have been available in the literature [15,16,17], there are few articles on the structural properties and antioxidant activity of the isolated lignins obtained from the sequential two-step formosolv fractionation using 88% FA followed by 70% aqueous FA solution. *Phragmites australis* (common reed), one of the most extensively distributed emergent plant species in the world, grows well in wetland environments, and it is rich in Dongting Lake, Hunan province. The potential productivity of *P. australis* biomass is up to 900,000 tons per year. It is very important and urgent to volarize *P. australis* biomass into renewable and sustainable chemistry. Therefore, this present work aims to fractionate lignin from *P. australis* using two-step formosolv fractionation processes, 88% FA as solvent followed by 70% aqueous FA. The obtained lignin fractions are elucidated by gel permeation chromatography (GPC), Fourier transform infrared spectroscopy (FTIR), thermogravimetric analysis (TGA), and two-dimensional heteronuclear single quantum coherence (2D-HSQC) nuclear magnetic resonance (NMR). The aromatic monomers of depolymerized lignin bio-oils are also qualitatively determined by gas chromatography-mass spectroscopy (GC/MS). Furthermore, the antioxidant capabilities of the obtained lignin samples are monitored with 1,1-diphenyl-2-picrylhydrazyl (DPPH) and 2′-azinobis-(3-ethylbenzthiazoline-6-sulphonate) (ABTS•^+^)radical scavenging activity using the UV-Vis spectroscopy method. Based on the experimental data, the objective of this study is to explore the changes in lignin structure and its antioxidant activity through two-step formosolv fractionation.

## 2. Experimental Methods

### 2.1. Isolation of Lignin through Two-Step Formosolv Fractionation

Lignin samples were isolated from *P. australis* by two-step formosolv fractionation using 88% FA followed by 70% aqueous FA solution, and the scheme of the procedure is shown in Figure 1.

*P**. australis* biomass was kindly gifted by Hunan Agricultural Science Academy (Changsha, China). The raw biomass was first crushed and screened with 40 mesh. A total of 500 g of the powder material was infiltrated with 88% FA with a total volume of 5000 mL. After reacting at 100 °C and 280 rpm for 3 h, the mixture was filtrated, and the residue was washed with extract solvent three times. The liquor from the filtration and washing was collected and evaporated to recover FA, the recovery efficiency of which was over 90% in our experiments. When the total volume of liquor was approx. 50 mL, 500 mL of deionic water were added to precipitate the lignin. Through centrifugation, the lignin precipitant was obtained as a solid residue. This as-obtained lignin was washed with deionic water and dried at 80 °C to achieve an FL-88% fraction with a yield of approx. 39 g, and the lignin content in FL-88% was calculated to be ca. 94%. Subsequently, 5 g of the resultant FL-88% fraction was dissolved in 500 mL of the 70% aqueous FA solution. The mixture was stirred at 100 °C and 280 rpm for 3 h. After the reaction was complete, both the supernatant and insoluble lignin precipitate were separated by centrifugation; the insoluble lignin was washed with deionic water to remove FA until the solution was neutral (pH ≈ 7.0), and approx. 2.46 g of FIL-70% fraction with the lignin content of ca. 96% was obtained after drying in an oven at 80 °C. The supernatant (e.g., soluble lignin fraction) was treated in two ways. In one way, the supernatant was rotary evaporated to approx. 50 mL and 500 mL deionic water was added to precipitate the lignin. After centrifugation and drying, approx. 2.50 g of FSL-70% fraction, with a lignin content of ca. 98%, was obtained. In another way, the supernatant was extracted with ten-fold ethyl acetate (EtAc), the extract was evaporated to recover EtAc. After drying, approx. 2.13 g of FL-EtAc fraction with a purity of ca. 98% was achieved. Mass balances of FL-88%, FIL-70%, FSL-70%, and FL-EtAc lignin fractions are listed in Table 1.

### 2.2. Structural Characteristics of the Isolated Lignin Fractions

#### 2.2.1. GPC Analysis

Due to poor solubility, lignin samples were first acetylated with acetic anhydride before GPC analysis [18]. GPC with a PLgel 15 μm MIXED-E column (Agilent 1260, Ramsey, MN, USA) was conducted to measure lignin molecular weight-average (M_w_), molecular number-average (M_n_), and polydispersity index (PDI). Tetrahydrofuran (THF) was used as the mobile phase with a flow rate of 1 mL·min^−^^1^. Polystyrene standards (Sigma-Aldrich, Saint Louis, MO, USA) with different molecular weights served as the internal standard.

#### 2.2.2. FTIR Analysis

FTIR (BRUKER, Bremen, Germany) was conducted to monitor the variance of functional groups in the isolated lignin fractions. The lignin sample and potassium bromide (KBr) were uniformly mixed with a mass ratio of 1:10. The mixture was then pressed into tablets, which were tested by FTIR. The FTIR spectra were performed in the frequency ranges from 4000 to 400 cm^−1^. Each spectrum was recorded over 10 scans, and the resolution was 0.5 cm^−1^.

#### 2.2.3. HSQC-NMR Analysis

HSQC-NMR (400 MHz, Bruker, Germany) was performed at 25 °C to provide accurate detailed structural information of the isolated lignin samples [19]. Briefly, 60 mg of the acetylated lignin were dissolved in 500 μL of dimethyl sulfoxide (DMSO-d6) and placed in a 5 mm NMR tube for NMR spectrum analysis. The NMR assay conditions were: The number of acquisition points was 2048 for 1TD and 256 for 2TD. The spectrum width (SW) was 15.9794 ppm with 1SW and 2SW values of 259.9935 and 15.9794 ppm, respectively. The scanning time (NS) was 64. The center frequencies were 4.7 ppm for O1P and 100 ppm for O2P.

#### 2.2.4. GC/MS Analysis

The aromatic monomers of FL-EtAc fraction were qualitatively determined on a Trace ISQ GC/MS equipped with a 30 m TR-WAXMS capillary column (30 m × 0.25 mm × 0.25 μm) (Thermo Fisher Scientific, Waltham, MA, USA). Briefly, the lignin sample was dissolved in EtAc solution and filtered by 0.22 µm membrane for GC/MS analysis. The GC conditions were: The split injection was 1:10. The injector temperature was set at 250 °C. The gas flow rate was fixed at 1 mL·min^−1^. A temperature program was applied as follows: 40 °C for 1 min, heating to 250 °C with the heating rate of 15 °C·min^−1^ and remaining at 250 °C for 10 min. The MS conditions were: EI source-detector, ion-source temperature 250 °C. MS scans from 33 to 500 amu. Compounds were identified using the ChemStation software and the replib library.

#### 2.2.5. TGA Analysis

The thermostability of the lignin fractions was carried out on a TGAQ50 thermal gravimetric analyzer (TA instruments, New Castle, DE, USA). Briefly, approx. 5 mg of the lignin samples were dried in a vacuum at 40 °C for 48 h. Then, the dried samples were analyzed by TGA under a nitrogen atmosphere from 20 to 800 °C. The heating rate was set at 10 °C·min^−1^.

### 2.3. Antioxidant Activity of the Isolated Lignin Fraction

#### 2.3.1. ABTS•^+^ Radical Scavenging Assay

The ABTS assay was performed at 734 nm on a UV-Vis spectroscopy (Youke Co., Shanghai, China) according to the previously reported method with minor modifications [20,21]. In brief, 7.4 mM ABTS stock solution and 2.6 mM potassium persulfate solution were mixed to generate ABTS radical cation (ABTS•^+^) after standing for 12–16 h at 25 °C in a dark environment. The as-obtained ABTS•^+^ solution was diluted with ethanol to an absorbance of 0.70 ± 0.02 at 734 nm. The lignin sample was dissolved in 90% dioxane/H_2_O mixture solution (*v*/*v*) to prepare different concentrations of 0.1 to 0.5 mg·mL^−1^. A total of 40 μL aliquot of sample was mixed with 4 mL of the ABTS•^+^ solution, and the mixture was incubated for 30 min in the dark for UV/VIS measurement at 734 nm, using ethanol solution as the blank control. All measurements were performed in triplicates. The ABTS•^+^ radical scavenging activity was calculated by Equation (1) using commercial antioxidant, butylated hydroxytoluene (BHT), as the positive control. The calculated IC_50_ value of ABTS•^+^ analysis represents the lignin concentration inhibiting 50% of the initial ABTS•^+^ radical.
(1)$$\mathrm{ABTs}{•}^{+}radical\;scavenging\;activity\;(\%)=\frac{A_{control}-A_{sample}}{A_{control}}\times 100\%$$
where *A_control_* and *A_sample_* are the absorbance at 734 nm of the control and sample, respectively.

#### 2.3.2. DPPH Radical Scavenging Activity Assay

The DPPH assay was measured at 517 nm on a UV-Vis spectroscopy (Youke Co., Shanghai, China) according to the method reported by Wei et al. [21]. The lignin sample was dissolved in 90% aqueous dioxane solvent to prepare lignin solutions with concentrations ranging from 0.1 to 0.5 mg·mL^−1^. DPPH solution with a concentration of 0.6 mmol·L^−1^ was prepared in ethanol. It was stored at 4 °C in the dark for use. Subsequently, the lignin sample solution (200 µL) was mixed with DPPH solution (4.3 mL), and the mixture was incubated for 30 min at 25 °C in the dark. The absorbance of the solutions was measured at 517 nm using ethanol solution as the blank control and BHT as the positive control. All measurements were performed in triplicates. The DPPH radical scavenging activity was calculated as Equation (2). The calculated IC_50_ value of DPPH analysis represents the lignin concentration inhibiting 50% of the initial DPPH radical.
(2)$$\mathrm{DPPH}\;radical\;scavenging\;activity\;(\%)=\frac{A_{{}_{Control}}-A_{Sample}}{A_{Control}}\times 100\%$$
where *A_Control_* and *A_Ssample_* are the absorbances at 517 nm of the control and sample, respectively.

## 3. Results and Discussion

### 3.1. Molecular Weights of Isolated Lignin Fractions

It has been found that the antioxidant capability of lignin macromolecules is greatly dependent on lignin reactive functional groups and heterogeneity [21]. Herein, GPC was conducted to determine the molecular weight of isolated lignin fractions during two-step formosolv fractionation and calculate the values of molecular weight-average (M_w_), molecular number-average (M_n_), and polydispersity index (PDI). The results are shown in Table 2.

As shown in Table 2, FIL-70% shows the highest values of M_n_, M_w_, and PDI, while FL-EtAc has the lowest values. For FIL-70% obtained as a precipitant in the 70% aqueous FA fractionation, the condensation of lignin is probably the major reaction, leading to a sharp increase in Mw value up to 14,314 Da. It was demonstrated that lignin condensation under acidic conditions mainly occurred between the nucleophilic benzylic carbocations (Cα) and the electrophilic aromatic ring [22]. However, FSL-70% and FL-EtAc show relatively lower Mn and Mw with respect to FL-88%. Moreover, FL-EtAc has a relatively smaller PDI than FL-88%. When lignin molecules dissolve in aqueous FA and EtAc solutions, the depolymerization reaction occurs to produce low-molecular-mass lignin monomers. Lignin with aromatic monomers and good homogeneity is of importance for its antioxidant activity [21]. Therefore, formosolv fractionation can lead to two competitive reactions of depolymerization and condensation of lignin. This phenomenon has been widely confirmed by many previously reported articles in the literature [22,23,24].

### 3.2. FTIR Analysis of Isolated Lignin Fractions

In order to shed light on the structural properties of the isolated lignin fractions during two-step formosolv fractionation, FTIR was used to unravel the functional groups’ information of the lignin samples. The results of FTIR curves are depicted in Figure 2.

As observed in Figure 2, the peak at 3436 cm^−1^ is assigned to the O–H stretching vibration in aromatic and aliphatic –OH groups of lignin [21]. It is clearly found that the absorption intensity of –OH group absorption peaks are slightly different for the isolated lignin samples. The signals at 2938 and 2838 cm^−1^ are related to the C–H asymmetric and symmetrical vibrations of methoxyl groups in lignin, and this absorption intensity of FL-EtAc was much stronger than the other three lignin samples, suggesting that FL-EtAc might have a higher content of methoxyl groups [25], which is contributed to antioxidant activity [21]. The peak at 1724 cm^−1^ is assigned to C=O stretching vibration of the lignin structure. In comparison with FL-88%, the absorbance intensity at 1724 cm^−1^ of the other three lignin samples, FSL-70%, FIL-70% and FL-EtAc, shows an upward trend, which might be due to another formylation reaction during 70% aqueous formosolv fractionation [9,13,17,22,24]. The isolated lignin samples show intensive peaks at 1606, 1513, and 1427 cm^−1^, which are typical representative signals of lignin aromatic skeletal vibrations [23]. The peak absorbance at 1221 cm^−1^ is assigned to C–C plus C–O plus C=O stretching of the condensed guaiacol (G_condensed_). Its intensity for the isolated lignin samples is decreasing in the order FIL-70% > FSL-70% > FL-88% > FL-EtAc. It indicates that the lignin condensation reaction in FIL-70% is superior to FSL-70%, FL-88%, and FL-EtAc. This hypothesis is in good conformity with the data of GPC analysis, where the Mw of FIL-70% is the highest, then FSL-70% and FL-88% and the Mw of FL-EtAc is the smallest (Table 1). The absorbance peaks at 1327, 1267 and 1221 cm^−1^ are related to the syringyl (S) units. The peaks at 1119 and 1039 cm^−1^ are assigned to guaiacyl (G) units, and 1172 cm^−1^ is the *p*-hydroxyphenyl (H) unit. FTIR analyses show that lignin from *P**. australis* consists of G-S-H subunits in the skeletal structure, although the real structure of lignin from *P**. australis* is still unknown. On the other hand, FTIR analyses also show that two-step formosolv fractionation does not damage the lignin skeletal structure; it only changes the reactive functional groups, e.g., hydroxyl (–OH), methoxyl (–OCH_3_) and carbonyl (C=O), which are very important for the antioxidant activity of lignin.

### 3.3. NMR Analysis of Isolated Lignin Fractions

To further acquire more accurate information on lignin structure, the isolated lignin fractions were analyzed by HSQC-NMR to unravel the aromatic units and different interunit linkages in lignin. The HSQC-NMR spectra are shown in Figure 3. Referring to the published articles for lignin signals distribution [10,21,23], the main substructures of the isolated lignin are listed in Figure 4.

As observed in Figure 3, three main regions are clearly found in HSQC spectra for all lignin samples. Namely, ^13^C-^1^H correlations at approx. *δ*_C_/*δ*_H_ 0–50/0–2.5 ppm for the aliphatic region, approx. *δ*_C_/*δ*_H_ 50–100/2.5–6.0 ppm for the oxygenated aliphatic region and approx. *δ*_C_/*δ*_H_ 100–170/6.0–10.0 ppm for the aromatic region [10,26]. In the aliphatic region, the signals at *δ*_C_/*δ*_H_15–30/0.5–2.5 ppm are commonly attributed to various types of fatty acids [10], and these structures are relatively stable during two-step formosolv fractionation processes. In the oxygenated aliphatic region, the signals at *δ*_C_/*δ*_H_ 55.5/3.7 ppm are associated with methoxyl (–OCH_3_) groups of lignin. The signal intensity of the methoxyl group for FL-EtAc is higher than the other three lignin samples. This phenomenon agrees well with the FTIR analysis in Figure 2. The correlations at *δ*_C_/*δ*_H_ 63.0/4.4 ppm related to β-*O*-4 units esterified in γ-positions (A^’^_γ_) are observed in all isolated lignins [27]. It is further demonstrated that the esterification reaction partially occurs between the aliphatic hydroxyl groups of lignin and FA during formosolv fractionation [4,9,10].

In the aromatic region, signals at δ_C_/δ_H_ 103.7/6.7 ppm are associated with the C_2,6_–H_2,6_ correlation of S and *δ*_C_/*δ*_H_ 106.0/7.2 ppm with Cα-oxidized S (S′) units are obviously observed. The correlation signals of the C_2_–H_2_, C_5_–H_5_ and C_6_–H_6_ of G units are found at *δ*_C_/*δ*_H_ 110.7/6.9, 114.8/6.7 and 118.9/6.8 ppm, respectively [10]. Further, signals at *δ*_C_/*δ*_H_ 131.2/7.6 ppm to C_2,6_–H_2,6_ in *p*-hydroxybenzoate substructures (PB) and *δ*_C_/*δ*_H_ 128.2/7.2 ppm associated with *p*-hydroxyphenyl (H) units are markedly observed from HSQC-NMR spectra [10]. A reasonable conclusion has been drawn from 2D HSQC spectra results that lignin from *P**. australis* consists of G/S/H subunits, which is in good accordance with the FTIR analysis (Figure 2). Furthermore, the marked signals at *δ*_C_/*δ*_H_ 106.4/6.6, 112.0/6.7 and 120.5/6.6 ppm are assigned to the C_2,6_-position of condensed S units and the C_2_ and C_6_-position of condensed G units, respectively. This phenomenon indicates that the lignin condensation reaction occurs during the formosolv fractionation process. The cross-signal of formate ester is markedly observed at *δ*_C_/*δ*_H_ 161.9/8.2 ppm, which might be attributed to the formylation reaction between lignin and FA [4,9,10].

### 3.4. GC/MS Analysis of Aromatic Monomers

To figure out the aromatic monomers, a standard GC/MS analysis of the FL-EtAc soluble in EtAc solution was performed. The GC chromatogram in Figure 5 reveals that a total of eight meaningful aromatic monomers, either S, G or H type units, are identified in the FL-EtAc fraction, such as 2-methoxy-4-vinylphenol, 2-methyl-benzaldehyde, vanillin, dibutyl phthalate, *p*-hydroxy benzaldehyde, 1-(4-hydroxy-3-methoxyphenyl)-ethanone or 4-hydroxy-3,5-dimethoxy-benzaldehyde, 1-(4-hydroxy-3,5-dimethoxy phenyl)-ethanone. It is worthily noticed that these compounds were formed by lignin depolymerization, mostly through FA-induced β-*O*-4 cleavage through a formylation, elimination and hydrogenolysis mechanism [11]. Under acidic conditions, proton-induced carbonium ion generation at the Cα position of the side chain could result in the homolytic breakage of the β-*O*-4 substructure [10,28]. The existence of these aromatic monomers in FL-EtAc is of great importance for antioxidant activity.

### 3.5. Proposed Mechanism of Depolymerization and Condensation of Lignin during Formosolv Fractionation

The combination of GPC, FTIR, NMR and GC/MS data, a proposed mechanism for the depolymerization and condensation of lignin during formosolv fractionation, is shown in Figure 6. For the condensation process, the proton-induced carbonium ion at the Cα position of the side chain could result in the homolytic cleavage of the β-*O*-4 bonds under acidic conditions. Simultaneously, the nucleophilic carbonium ion and electrophilic aromatic ring underwent a condensation reaction via carbon–carbon bond linkage, causing an increase in the residue lignin molecular weight. In the depolymerization process, the formylation reaction occurred between FA and the hydroxy group at the C_γ_/Cα position. Subsequently, the elimination of the formate at the C_γ_/Cα position through abstraction of the proton in the Cα/C_γ_ would lead to the formation of an unsaturated C=C bond. Finally, the cleavage of β-*O*-4 linkages broke via hydrolysis under acidic conditions to result in low-molecular-mass aromatic monomers. Previous works have demonstrated the importance of Cα/C_γ_ oxidation in promoting lignin depolymerization, which can result in a high yield of low-molecular-mass aromatics [4,12].

### 3.6. Thermal Stability of Isolated Lignin Fractions

TGA/DTG analysis was performed to characterize the thermal stability of the isolated lignin fractions derived from two-step formosolv fractionation; the results are shown in Figure 7.

As observed in Figure 7a, the initial weight loss of the lignin samples before 150 °C was attributed to water evaporation. The main weight loss stage occurred between 150 and 450 °C, during which inter-unit linkages of the lignins broke up, and monomeric phenols evaporated, causing the major weight-loss stage. Above 500 °C, the decomposition of the aromatic rings caused weight loss [29]. The residue mass of FL-EtAc fractions is approx. 26% after 800 °C, while the residue weight of the other three lignin fractions, FL-88%, FSL-70% and FIL-70%, is approx. 35–38% after 800 °C. As seen in Figure 7b, the maximum decomposition temperature (Tm) of the FL-EtAc is 369 °C, higher than that of FSL-70%, FIL-70% and FL-88%. The Tm value shifting to a higher temperature suggests that the FL-EtAc is more thermally stable than the other three lignin samples. It was demonstrated that the thermal stability of lignin was highly attributed to the content of β-*O*-4 linkages in the lignin polymers [10,30], the higher the thermal stability, the less content of the β-*O*-4 linkages was [10]. During the second 70% formosolv fractionation and EtAc extraction, some β-*O*-4 linkages in the lignin structure broke, leading to a decrease in the content of β-*O*-4 linkages in FL-EtAc. On the other hand, the second formylation modification during 70% formosolv fractionation could increase thermal stability in materials [31]. Jin and co-workers considered that the higher Tm value was mainly contributed to the higher purity of FL-EtAc lignin fraction, which showed a better positive effect on thermal stability [31]. Although their condensation degree and molecular weight are different, FSL-70% and FIL-70% endow many similar Tm values. Therefore, for the two-step formosolv fractionation process, the β-*O*-4 linkages cleavage and purity of lignin in the second 70% formosolv fractionation show some positive effect on the thermal stability of the isolated lignins.

### 3.7. Antioxidant Activity Analysis of Isolated Lignin Fractions

In order to evaluate the antioxidant capacity of the isolated lignin samples, ABTS•^+^ and DPPH assays were chosen to determine the radical scavenging activity using commercial antioxidants, with BHT as the positive control, and the results are shown in Figure 8.

As observed from Figure 8a, the ABTS•^+^ radical scavenging ability of isolated lignin samples increases with the concentration of lignin fractions, and FL-EtAc has the highest ABTS activity among the isolated lignin fractions, even equal to the commercially available antioxidant BHT at the same concentration of 2.0 mg·mL^−1^. Comparing the calculated IC_50_ value of ABTS•^+^ analysis of the isolated lignin fractions (Figure 8b), FL-88% has the lowest IC_50_ value in ABTS•^+^ analysis, 0.55 mg·mL^−1^, followed by FL-EtAc and FIL-70%, and FSL-70% has the highest IC_50_ value, 1.29 mg·mL^−1^. The statistical analysis (ANOVA) of the IC_50_ value in ABTS•^+^ analysis shows that there are significant differences between BHT, FSL-70% and other lignin samples. As expected, the DPPH radical scavenging ability of the isolated lignin samples also increases with the concentration of lignin fractions (Figure 8c). As seen in Figure 8c, the DPPH radical scavenging activity ranged from 55% to 75% at the concentration of 2.0 mg·mL^−1^ for the isolated lignin fractions, where the decrease was in the order of FL-EtAc > FL-88% > FlL-70% > FSL-70%. Correspondingly, FSL-70% has the highest calculated IC_50_ value of DPPH analysis (1.36 mg·mL^−1^) (Figure 8d). The statistical analysis (ANOVA) of the IC_50_ value in DPPH analysis shows that there are significant differences between FSL-70% and the other tested samples. The data show that two-step formosolv fractionation can produce specific lignin with different radical scavenging activities. According to previous reports, both phenolic hydroxyl groups and methoxyl groups in the lignin structure had a positive effect on the radical scavenging activity [21,32,33,34]. When phenolic –OH at the Cα position in the lignin structure was oxidized to α-C=O, the antioxidant ability of lignin dramatically decreased [3]. The direct correlation between the lignin structure and its antioxidant activity is under study in our lab.

## 4. Conclusions

Two-step formosolv fractionation of lignin from biomass can cause lignin depolymerization if accompanied by a condensation reaction, leading to different molecular weight and structural properties of the isolated lignin fractions. The structural properties of the obtained lignin fractions are systematically characterized by GPC, FTIR, HSQC-NMR and TGA techniques. The depolymerized lignin monomers are extracted by ethyl acetate and analyzed by GC/MS. At the same time, the lignin fractions exhibit different ABTS•^+^ and DPPH radical scavenging activity, which could be attributed to their structural properties, low molecular weight and polydispersity index. Overall, two-step formosolv fractionation can produce specific lignin fractions with different radical scavenging activities.

## Figures and Tables

**Figure 1 molecules-27-02905-f001:**
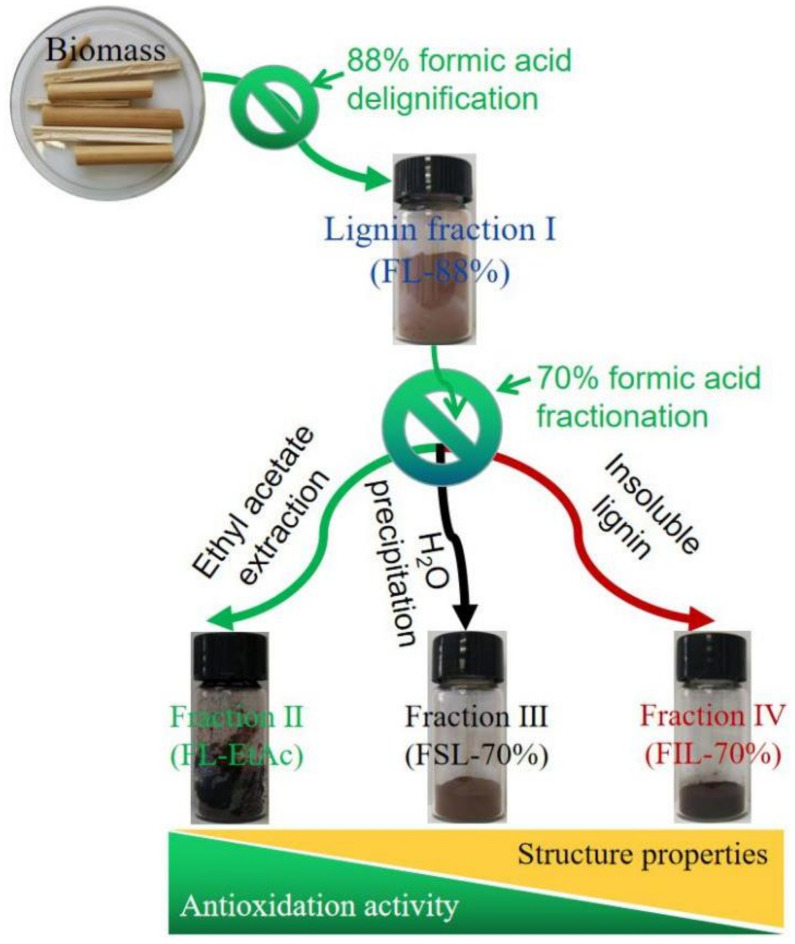
Scheme of lignin fractions preparation from *P**. australis* biomass by sequential two-step formosolv fractionation.

**Figure 2 molecules-27-02905-f002:**
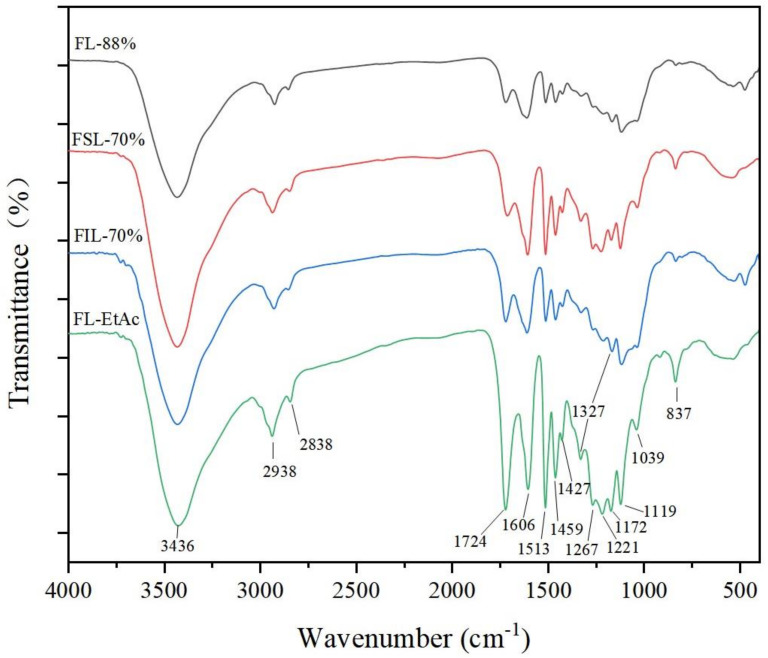
FTIR spectra of unacetylated lignin samples obtained from two-step formosolv fractionation.

**Figure 3 molecules-27-02905-f003:**
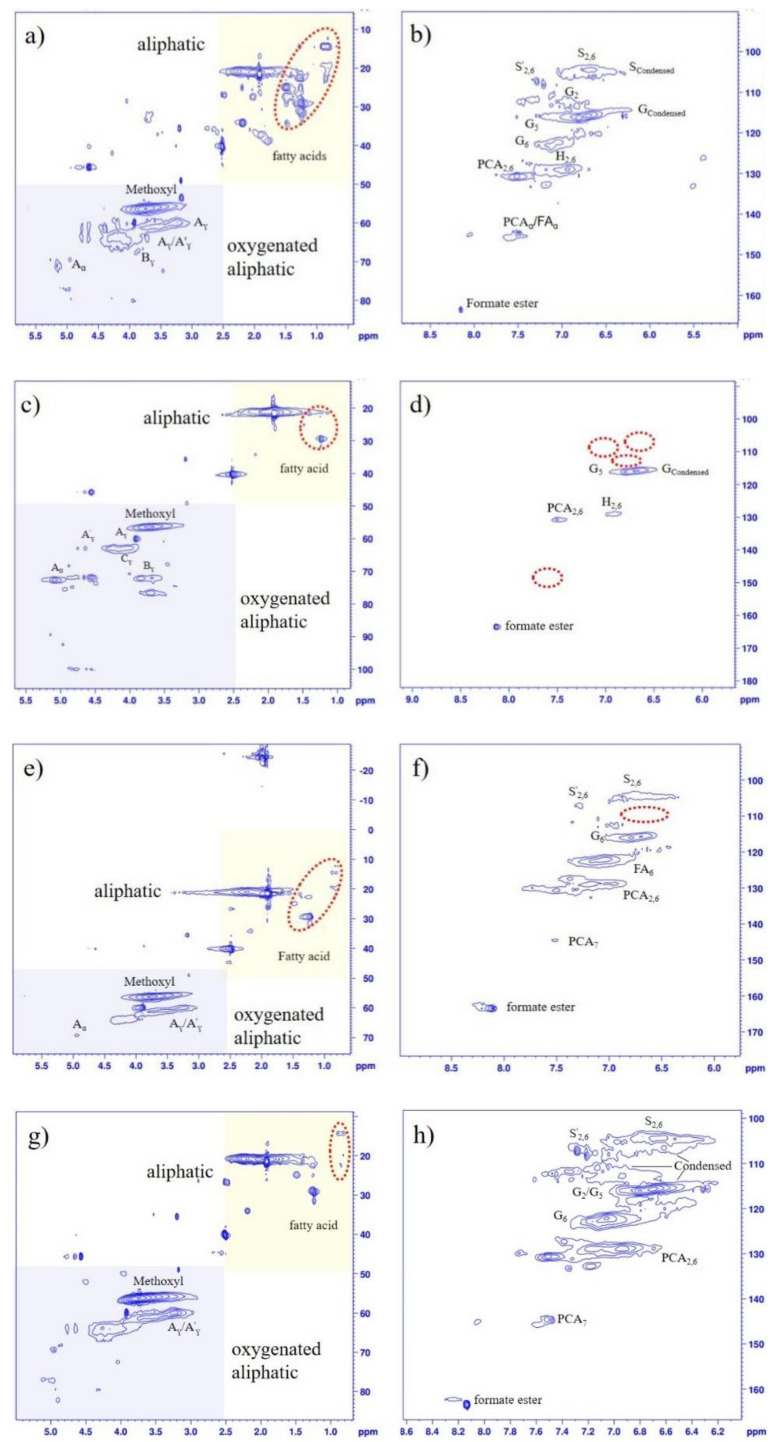
HSQC-NMR spectra of the isolated lignin fractions obtained from two-step formosolv fractionation (left column: side chain region; right column: aromatic region). (**a**,**b**): FL-88%; (**c**,**d**): FIL-70%; (**e**,**f**): FSL-70%; (**g**,**h**): FL-EtAc.

**Figure 4 molecules-27-02905-f004:**
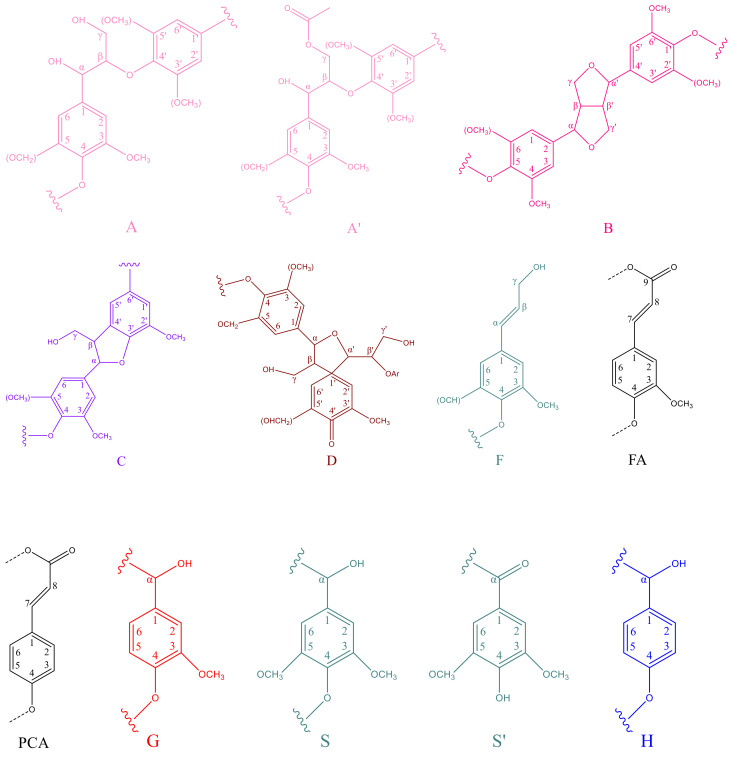
The main substructure of the isolated lignin analyzed by HSQC-NMR. A and A’ is different in the functional group in C_γ_ position. S and S’ is different in the functional group in C_α_ position.

**Figure 5 molecules-27-02905-f005:**
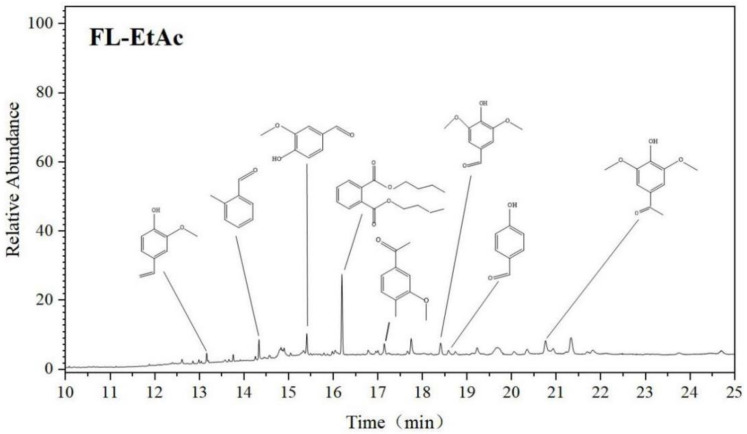
GC/MS analysis for the aromatic monomers in FL-EtAc fraction.

**Figure 6 molecules-27-02905-f006:**
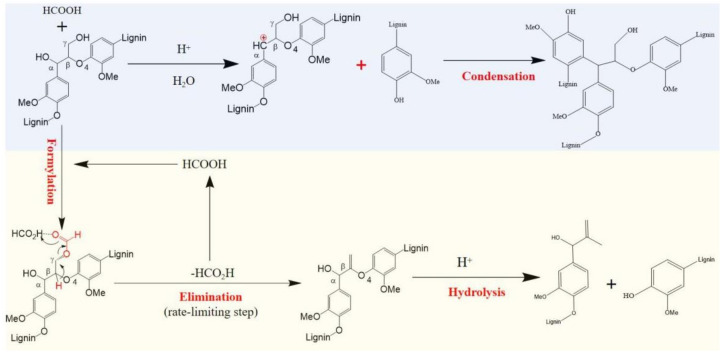
Proposed mechanism for the depolymerization and condensation of lignin during formosolv fractionation. Up: condensation scheme; Down: depolymerization scheme.

**Figure 7 molecules-27-02905-f007:**
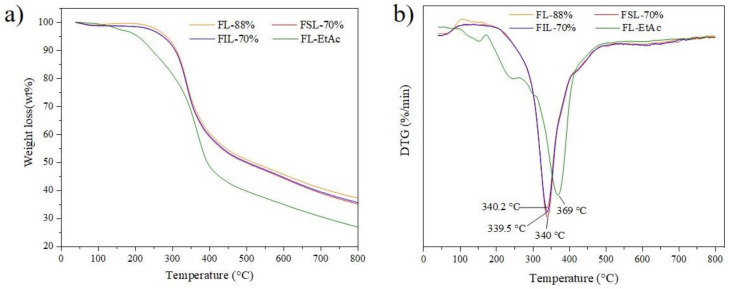
TGA/DTG analysis of lignin samples obtained from two-step formosolv fractionation. (**a**) TGA curves; (**b**) DTG curves.

**Figure 8 molecules-27-02905-f008:**
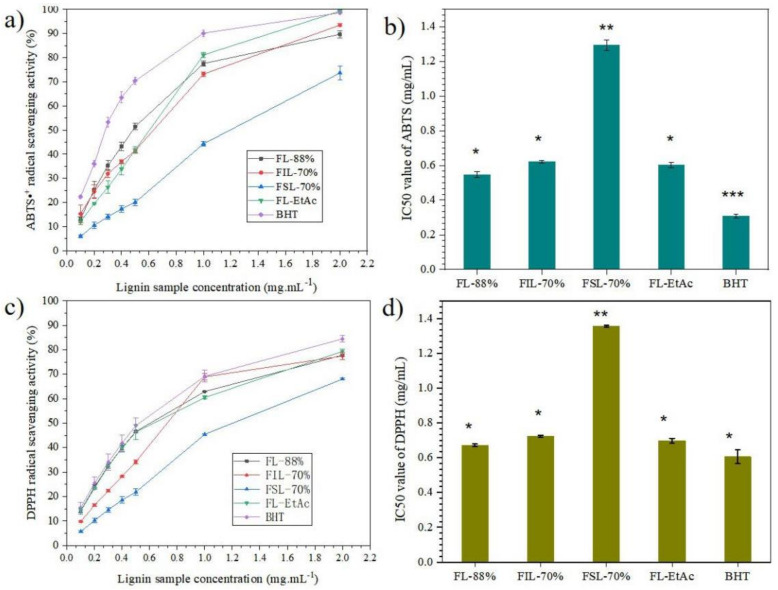
Antioxidant capacities of ABTS and DPPH radical scavenging activity for isolated lignin fractions. (**a**) ABTS radical scavenging activity; (**b**) IC_50_ of ABTS analysis; (**c**) DPPH radical scavenging activity; (**d**) IC_50_ of DPPH analysis. BHT was used as the positive control. Different asterisks represent significant differences (*p* < 0.01).

**Table 1 molecules-27-02905-t001:** Mass balances of FL-88%, FIL-70%, FSL-70%, and FL-EtAc lignin fractions.

FL-88% Fraction (g)	FIL-70% Fraction		FSL-70% Fraction ^a^		FL-EtAc Fraction ^b^	
	Mass (g)	Percentage (%)	Mass (g)	Percentage (%)	Mass (g)	Percentage (%)
5.0001	2.4625	49.25	2.498	49.96	2.1275	42.55

Notes: superscript letter a means the yield of lignin via the way of FL-88%, FIL-70%, and SL-70%; superscript letter b means the yield of lignin via the way of FL-88%, FIL-70%, and FL-EtAc.

**Table 2 molecules-27-02905-t002:** Molecular weight analyses of acetylated lignin samples by GPC.

Trial	Lignin Fraction	M_n_ (Da)	M_w_ (Da)	PDI
1	FL-88%	2758	7417	2.7
2	FSL-70%	3398	9270	2.7
3	FIL-70%	4108	14314	3.5
4	FL-EtAc	1876	2748	1.5

PDI: Polydispersity index.

## Data Availability

Not applicable.

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
