# Peer review of "Effect of Two-Step Formosolv Fractionation on the Structural Properties and Antioxidant Activity of Lignin"

_molecules, 2022, doi:10.3390/molecules27092905_

Round 1

Reviewer 1 Report

The authors have utilized a two-step formosolv fractionation process to produce different lignin fractions from a grassy biomass that exhibited varying level of antioxidant activity. There are minor corrections to be suggested, but otherwise the manuscript is well-written, scientifically sound, and may be accepted for publication.

1) Two different species are cited in the abstract and in the main text (Phragmites australis Vs. Triarrhena lutarioriparia L. Liu). Which species was actually used? Please correct.

Also, if applicable, please show pictures of actual (cut or chopped) biomass (in your lab) and the FL-88% lignin fraction (in Figure 1?).

2) In the introduction, please provide additional significance of the chosen biomass model system, such as annual availability and potential for cultivating on a large scale in the future. The current rationale is not enough to justify the utilization of this species.

3) Isn't the yield of FL-88% fraction too low at 7.8% (39 g out of 500 g)? What is the corresponding delignification efficiency? Please compare your yields with previous literature and justify your choice to continue fractionating this meager lignin source.

4) Looks like there is 26% (FSL-70%) to 32% (FL-EtAc) loss (of the FL-88% fraction) during the downstream fractionation of lignin. Couldn't you have simply evaporated/freeze dried the formic acid soluble fraction (FSL-70%) and later washed the recovered solids with water to improve the yield? By using water precipitation and EtAc extraction, the loss of lignin has increased.

5) There are minor spelling and grammatical corrections, please proof read the manuscript one more time.

For example, the conclusion could read as follows:

a) Line 379: It is systematically not symmetrically.

b) Line 384: can "produce lignin fractions with different radical scavenging activity".

c) Line 382: ...activity, which could be attributed to their...

Author Response

Point 1: Two different species are cited in the abstract and in the main text (Phragmites australis Vs. Triarrhena lutarioriparia L. Liu). Which species was actually used? Please correct.

Also, if applicable, please show pictures of actual (cut or chopped) biomass (in your lab) and the FL-88% lignin fraction (in Figure 1?).

Response 1: Thank you for your good comments. The Latin name of the used biomass is Phragmites australis, not Triarrhena lutarioriparia L. Liu. All revisions are marked in red throughout the manuscript.

Also, the pictures of atual biomass and lignin fractions in Figure 1 have been provided in the revised version.

Point 2: In the introduction, please provide additional significance of the chosen biomass model system, such as annual availability and potential for cultivating on a large scale in the future. The current rationale is not enough to justify the utilization of this species.

Response 2: Thank you for your good suggestions. The annual availability and potential cultuvation productivity of the chosen biomass have been provided in the revised version.

Point 3: Isn't the yield of FL-88% fraction too low at 7.8% (39 g out of 500 g)? What is the corresponding delignification efficiency? Please compare your yields with previous literature and justify your choice to continue fractionating this meager lignin source.

Response 3: Thank you for your good comments. The delignification efficiency of formic acid fractionation was reported in our previous work (Green Energy & Environment, 2022, 7, 172-183. https://doi.org/10.1016/j.gee.2020.08.006.), which value was 65% under the optimal conditions. However, the objective of this present work is to investigate the effect of two-step formosolv fractionation on the structural properties and antioxidation activity of lignin. Therefore, the delignification effeciency from biomass by formic acid fractionation is not the key issue in this present work.

Point 4: Looks like there is 26% (FSL-70%) to 32% (FL-EtAc) loss (of the FL-88% fraction) during the downstream fractionation of lignin. Couldn't you have simply evaporated/freeze dried the formic acid soluble fraction (FSL-70%) and later washed the recovered solids with water to improve the yield? By using water precipitation and EtAc extraction, the loss of lignin has increased.

Response 4: Thank you for your good comments. In fact, the loss of lignin for preparation of FSL-70% and FL-EtAc fractions is high in the present work. The reasonable explanation is attributed to the fact that lignin sample is sticked to the glass flask due to its high viscosity after drying, especially for FL-EtAc fraction. On the other hand, some lignin dissolving in formic acid solution also caused the loss of lignin. Therefore, by using water precipitatio and EtAc extraction, the loss of lignin is high. Becuase we are focusing on the  effect of two-step formosolv fractionation on the structural properties and antioxidation activity of lignin, we can ignore the loss of lignin in this present work. In future, we will increse the lignin yield through large-scale and considering the dissolving lignin, the precipited lignin and the lignin sticking to the flask.

Point 5: There are minor spelling and grammatical corrections, please proof read the manuscript one more time.

For example, the conclusion could read as follows:

  1. a) Line 379: It is systematically not symmetrically.
  2. b) Line 384: can "produce lignin fractions with different radical scavenging activity".
  3. c) Line 382: ...activity, which could be attributedto their...

Response 5: Thank you for your good comments. We have carefully checked the manuscript sentence by sentence to avoid the spelling and grammatical errors. All revisions are marked in red in the revised version.

Reviewer 2 Report

The manuscript submitted by the authors investigated the effect of two-step formic acid fractionation on the structural and antioxidant properties of lignin from a local native plant. The lignin obtained after second fractionation step was extracted further with ethyl acetate. The resulting fractions of lignin were thoroughly characterized for their structure using relevant chromatographic and spectroscopic methods. Their antioxidant activity was assessed using ABTS and DPPH assays and some revealed to be on par with a commercial standard.

The manuscript is relatively well-structured and readable and the methods are described decently to the standard of the journal. The characterization of the molecules was done in good standard methods of the field. However, the manuscript suffers greatly from weak story line, missing critical compositional information, control reference, several analytical details and overall relevant purpose, especially pointing towards the potential application that the authors tried to suggest. Additionally there are many grammatical and language styles error that can hamper readability. Hence, the suggestion by this reviewer for major revision.

Major comments

  1. Line 21-23 and Sub-chapter 3.7 (lines 347-374): It is not clear what is the difference implied between the results of antioxidant assays using ABTS and DPPH. These are just two compounds used to assess antioxidant activity, apart from other compounds and methods. The results and trends in terms of activity of the lignin fractions were even similar in both assays (Figure 8). The sentence in the abstract needs to be qualified, and depending on that, further explanation in the discussion on sub-chapter 3.7 might need expansion.
  2. Lines 65-68: the rationale of having 2-step fractionation needs to be elaborated. The fact that something has not been done does not simply justify its execution. There should be scientific and techno-economical reasoning behind it, especially correlation with antioxidant activity, which is the purpose of this work. What is the hypothesis?
  3. Lines 87-109: The mass balance is low on the solid basis (ca. 8% for FL-88% and ca. 2% for FSL-70%). Explanation must be given to this low result. Even better if there could be comparison to other previous work using the same biomass and/or pretreatment. A better alternative could also be to display it in the “Results and discussion” section.
  4. Lines 87-109: The mass balance should be expressed in terms of lignin content instead of solid basis (dry matter). Biomass composition analysis is an important information, especially when using native local plant that has not been widely studied. This is missing from the manuscript. Composition analysis (e.g. NREL method) should be done for the original biomass prior to fractionation and all the lignin fractions.
  5. Line 161-165: What was the solvent used to dissolve the lignin for ABTS assay? This was not clear from the text.
  6. Sub-chapters 3.1-3.6: In the characterization of lignin structure, which consists the bulk of the work, the inclusion of the original lignin material (or at least the milled wood lignin) as control is missing. This is critical, as many interpretations in the structural changes (notably on lignin condensation, which cannot be seen by molecular weight alone) cannot be fully assessed by the methods used, especially the 2D-HSQC NMR. The sub-chapter 3.5 is therefore mostly theoretical work instead of results based. A control reference or otherwise a strong justification why it was not included should be given.
  7. Figure 8 (line 352): Statistical test for significance (e.g. ANOVA) in the antioxidant assay results, at least on the IC50 is needed, to justify the order of magnitude of the fractions.
  8. Sub-chapter 3.7: Correlation between the structural changes or differences and lignin fractions is not clear. Elaboration on this aspect should be made.
  9. Sub-chapter 3.7 and conclusions: Since the antioxidant of lignin fraction after the first formosolv treatment did not seem to differ much than the ethyl acetate treatment of the second step (Figure 8), it should be indicated if this 2-step fractionation has any added techno-economical potential benefit compared to the single step.

Minor comments

  1. The title (lines 2-4) is too long to be necessary, some details, e.g. “using 88% formic acid followed by…” can actually be skipped.
  2. The biomass mentioned in the abstract (line 13) is different than the one mentioned in the rest of the manuscript.
  3. Although it is known that formic acid has be used to fractionate lignin, the rationale should be given to justify the use of the otherwise toxic solvent, i.e. in the introduction by expanding e.g. those in lines 43-44. Why use formic acid instead of other solvents? What are the supposed benefits? This is not a crucial comment, but it will be good to improve.
  4. The correct terms should be “antioxidant” instead of “antioxidation” (e.g. lines 3, 27, 33, 68, 96, 108, 353, 371, 373, 374) and “structural” instead of structure (e.g. lines 68, 96, 108).
  5. Grammatical and language style errors (lines 21, 22, 31, 39, 40, 59, 67, 98, 156-157, 197, 213, 264, 345, 348, 353).
  6. Wrong unit for the wavenumber (line 209).
  7. Reference(s) for the assignment of the FTIR spectra is/are lacking (line 239).
  8. Figure 8 (line 352): the values in the scales used both for antioxidant activity and the IC50 should be the same for easier comparison. The DPPH antioxidant activity ends at 90% instead of 100% in ABTS and the DPPH IC50 starts at 0.5 mg/mL instead of 0% in ABTS.

Author Response

Response to Reviewer 2 Comments

Major comments

Point 1: Line 21-23 and Sub-chapter 3.7 (lines 347-374): It is not clear what is the difference implied between the results of antioxidant assays using ABTS and DPPH. These are just two compounds used to assess antioxidant activity, apart from other compounds and methods. The results and trends in terms of activity of the lignin fractions were even similar in both assays (Figure 8). The sentence in the abstract needs to be qualified, and depending on that, further explanation in the discussion on sub-chapter 3.7 might need expansion.

Response 1: Thank you for your good comments. There are four common methods to assess the antioxidant activity of lignin, including DPPH free radical scavenging ability, ABTS•+ , free radical scavenging ability, •OH free radical scavenging ability and reducing power. In our present work, ABTS•+ and DPPH free radical scavenging ability were used to evaluate different lignin fractions’ antioxidant activity. As seen in Figure 8, the free radical scavenging assay of lignin fractions is related to the concentration of lignin. As the lignin concentration increases, the free radical scavenging effect also increases. Generally, the antioxidant activity of lignin is related to its structural properties, phenolic hydroxyl groups, methoxyle groups and heterogeneity. As the comments from reviewer, the sentence in the abstract has been qualified and further explanation in the discussion is also expansed. All revisions are marked in red in the revised version.

Point 2: Lines 65-68: the rationale of having 2-step fractionation needs to be elaborated. The fact that something has not been done does not simply justify its execution. There should be scientific and techno-economical reasoning behind it, especially correlation with antioxidant activity, which is the purpose of this work. What is the hypothesis?

Response 2: Thank you for your good comments. Numerous previous works have demonstrated that lignin can be depolymerized in formic acid solution in combination with condensation reaction. Due to different solubility of lignin in different concentration of formic acid, a reasonable hypothesis has been proposed that the tailor-made lignin fractions from formic acid fractionation will show different applications related to their structural properties. Therefore, the objective of this present work is to investigate the effect of two-step formosolv fractionation on the structural properties and antioxidation activity of lignin using 88% formic acid followed by 70% aqueous formic acid solvent.

.

Point 3: Lines 87-109: The mass balance is low on the solid basis (ca. 8% for FL-88% and ca. 2% for FSL-70%). Explanation must be given to this low result. Even better if there could be comparison to other previous work using the same biomass and/or pretreatment. A better alternative could also be to display it in the “Results and discussion” section.

Response 3: Thank you for your good comments. The delignification efficiency of formic acid fractionation was reported in our previous work (Green Energy & Environment, 2022, 7, 172-183. https://doi.org/10.1016/j.gee.2020.08.006.), which value was 65% under the optimal conditions. In fact, the yield of FL-88% and FSL-70% is very low in the present work. The reasonable explanation is attributed to the fact that lignin sample is sticked to the glass flask due to its high viscosity after drying. On the other hand, some lignin dissolving in formic acid solution also caused the loss of lignin. In future, we will increse the lignin yield through large-scale and considering the dissolving lignin, the precipited lignin and the lignin sticking to the flask. Simultaneously, we will explore the mass balance of formic acid fractionation and compare with other pretreatment. All in all, the objective of this present work is to investigate the effect of two-step formosolv fractionation on the structural properties and antioxidation activity of lignin. Therefore, the mass balance of formic acid fractionation is not the key issue in this present work. Basic mass balance of two-step formic acid fractionation has been provided in the revised version.

Point 4: Lines 87-109: The mass balance should be expressed in terms of lignin content instead of solid basis (dry matter). Biomass composition analysis is an important information, especially when using native local plant that has not been widely studied. This is missing from the manuscript. Composition analysis (e.g. NREL method) should be done for the original biomass prior to fractionation and all the lignin fractions.

Response 4: Thank you for your good comments. The content of lignin fractions obtained from two-step formic acid fractionation is determined and provided in the revised version.

Point 5: Line 161-165: What was the solvent used to dissolve the lignin for ABTS assay? This was not clear from the text.

Response 5: Thank you for your good comments. 90% of dioxane/H2O (v/v) was used to dissolve the lignin for ABTS and DPPH assay in our present work. All revsions are added in the revsied version.

Point 6: Sub-chapters 3.1-3.6: In the characterization of lignin structure, which consists the bulk of the work, the inclusion of the original lignin material (or at least the milled wood lignin) as control is missing. This is critical, as many interpretations in the structural changes (notably on lignin condensation, which cannot be seen by molecular weight alone) cannot be fully assessed by the methods used, especially the 2D-HSQC NMR. The sub-chapter 3.5 is therefore mostly theoretical work instead of results based. A control reference or otherwise a strong justification why it was not included should be given.

Response 6: Thank you for your good comments. The lignin real structure of biomass is unknown up to now because of the complexity of lignin structure. Whatever method used for lignin preparation including ballmilled wood lignin should damage lignin structure. In our present work, FL-88% was employed as reference to examine the effect of 70% formic acid fractionation on the changes of lignin fractions structure and antioxidatant activity. Therefore, milled wood lignin is not used as control in our work. In future, the effect of preparation procedures of lignin, including ballmill, ionic liquids, diluted acid, and DES, on the structural properties and activity of the talored-made lignin will be under considered in our lab.

Points 7: Figure 8 (line 352): Statistical test for significance (e.g. ANOVA) in the antioxidant assay results, at least on the IC50 is needed, to justify the order of magnitude of the fractions.

Response 7: Thank you for your good comments. ANOVA analysis in the antioxidant assay results, including IC50 values has been added in Figure 8 in the revised version.

Points 8: Sub-chapter 3.7: Correlation between the structural changes or differences and lignin fractions is not clear. Elaboration on this aspect should be made.

Response 8: Thank you for your good comments. The structural changes or differences of lignin fractions obtained from two-step formic acid fractionation are elaborated in the revised version, including GPC, FTIR, NMR and TGA results.

Points 9: Sub-chapter 3.7 and conclusions: Since the antioxidant of lignin fraction after the first formosolv treatment did not seem to differ much than the ethyl acetate treatment of the second step (Figure 8), it should be indicated if this 2-step fractionation has any added techno-economical potential benefit compared to the single step.

Response 9: Thank you for your good comments. From structural properties of FL-88% and FL-EtAc fractions, depolymerization and condensation reaction are observed. Furthermore, some aromatic monomers were determined in FL-EtAc by GC/MS. However, the antioxidant activity of both lignin fractions did not seem to differ much. It indicates that the effect mechanism of antioxidant activity of lignin is greatly complex. In future, we will focus on much efforts to highlight the antioxidant activity of lignin. Also, the techno-economical potential benefit should be compared between single formic acid treatment and two-step formic acid treatment.

Minor comments

Point 1: The title (lines 2-4) is too long to be necessary, some details, e.g. “using 88% formic acid followed by…” can actually be skipped.

Response 1: Thank you for your good comments. The title is corrected and the words of “using 88% formic acid followed by…” are deleted.

Point 2: The biomass mentioned in the abstract (line 13) is different than the one mentioned in the rest of the manuscript.

Response 2: Thank you for your good comments. The Latin name of biomass is corrected to “Phragmites australisthroughout the manuscript.

Point 3: Although it is known that formic acid has be used to fractionate lignin, the rationale should be given to justify the use of the otherwise toxic solvent, i.e. in the introduction by expanding e.g. those in lines 43-44. Why use formic acid instead of other solvents? What are the supposed benefits? This is not a crucial comment, but it will be good to improve.

Response 3: Thank you for your good comments. The merits using formic acid as fractionation solution have been added in the revised version.

Point 4: The correct terms should be “antioxidant” instead of “antioxidation” (e.g. lines 3, 27, 33, 68, 96, 108, 353, 371, 373, 374) and “structural” instead of “structure” (e.g. lines 68, 96, 108).

Response 3: Thank you for your good comments. All these error terms have been corrected in the revised version.

Point 5: Grammatical and language style errors (lines 21, 22, 31, 39, 40, 59, 67, 98, 156-157, 197, 213, 264, 345, 348, 353).

Response 5: Thank you for your good comments. We have carefully checked the mansucript and corrected the grammatical and language style errors in the text.

Point 6: Wrong unit for the wavenumber (line 209).

Response 6: Thank you for your good comments. The unit for the wavenumber is corrected to cm-1 in the revised version.

Point 7: Reference(s) for the assignment of the FTIR spectra is/are lacking (line 239).

Response 7: Thank you for your good comments. The reference for the assignment of the FTIR is added in the revised version.

Point 8: Figure 8 (line 352): the values in the scales used both for antioxidant activity and the IC50 should be the same for easier comparison. The DPPH antioxidant activity ends at 90% instead of 100% in ABTS and the DPPH IC50 starts at 0.5 mg/mL instead of 0% in ABTS.

Response 8: Thank you for your good comments.The values in the scales used both for antioxidant activity and the IC50 (Figure 8) have been unifomed in the revised version.

Round 2

Reviewer 2 Report

The manuscript has been well-improved, however there are still several things to clarify and correct

  1. If the intended biomass is Phragmites australis, then it is not a local native plant as it is widely if not globally distributed invasive species. Several statements (e.g. lines 70-72) need to be qualified.
  2. The term purity (i.e. lines 110, 113, 116) was not clearly defined.
  3. The mass balance mentioned in the revision was not provided.
  4. The statistical analysis (ANOVA) mentioned in the revision was not provided either in the methods or in the results (i.e. Figure 8).
  5. There are still typos and errors (e.g. lines 21, 381, 386).

Author Response

Point 1: If the intended biomass is Phragmites australis, then it is not a local native plant as it is widely if not globally distributed invasive species. Several statements (e.g. lines 70-72) need to be qualified.

Response 1: Thank you for your good comments. The statements in lines 70-72 have been revised. Phragmites australis (common reed), one of the most extensively distributed emergent plant species in the world, grows well in wetland environment. 

Point 2: The term purity (i.e. lines 110, 113, 116) was not clearly defined.

Response 2: Thank you for your good comments. The purity in lines 110, 113, 116 is defined to the lignin content. All revisions are corrected and marked in red in the text.

Point 3: The mass balance mentioned in the revision was not provided.

Response 3: Thank you for your good comments. The mass balance is listed as follows and provided in the revised version.

Table 1. Mass balance of FL-88%,FIL-70%, FSL-70% and FL-EtAc lignin fractions

FL-88% fraction (g)

FIL-70% fraction

FSL-70% fraction

FL-EtAc fraction

Mass (g)

Percentage (%)

Mass (g)

Percentage (%)

Mass (g)

Percentage (%)

5.0001

2.4625

49.25

2.498

49.96

2.1275

42.55

Point 4: The statistical analysis (ANOVA) mentioned in the revision was not provided either in the methods or in the results (i.e. Figure 8).

Response 3: Thank you for your good comments. ANOVA has been provided in Figure 8. Different asterisk in Figure 8 represents significant difference (p<0.01).

Point 5: There are still typos and errors (e.g. lines 21, 381, 386).

Response 5: Thank you for your good comments.  All typos and errors in lines 21,381, 386 are all corrected. For instance, “ specif” in line 21 has been corrected to “specific”. “oder” in line 381 has been corrected to “order”. “antioxidatant” in line 386 has been corrected to “antioxidant”.

This manuscript is a resubmission of an earlier submission. The following is a list of the peer review reports and author responses from that submission.